# Diagnosis, Treatment, and Outcome of Coccidioidal Osseous Infections: A Systematic Review

**DOI:** 10.3390/jof10040270

**Published:** 2024-04-05

**Authors:** Andreas G. Tsantes, Christos Koutserimpas, Symeon Naoum, Lida-Paraskevi Drosopoulou, Ellada Papadogeorgou, Vasileios Petrakis, Kalliopi Alpantaki, George Samonis, Enejd Veizi, Dimitrios V. Papadopoulos

**Affiliations:** 1Laboratory of Hematology and Blood Bank Unit, “Attikon” University Hospital, School of Medicine, National and Kapodistrian University of Athens, 12462 Athens, Greece; 2Microbiology Department, “Saint Savvas” Oncology Hospital, 11522 Athens, Greece; ledadrossos@yahoo.gr; 3Orthopaedic Surgery and Sports Medicine Department, Croix-Rousse Hospital, University Hospital, 69317 Lyon, France; chrisku91@hotmail.com; 4Department of Anatomy, School of Medicine, Faculty of Health Sciences, National and Kapodistrian University of Athens, 75 Mikras Asias Str., Goudi, 11527 Athens, Greece; 5Department of Trauma and Orthopedics, Royal Berkshire Hospital, Reading RG1 5AN, UK; naoumsimeon@gmail.com; 6Department of Orthopedics, Interbalkan Medical Center, 55535 Thessaloniki, Greece; ellada955@gmail.com; 72nd University Department of Internal Medicine, University General Hospital of Alexandroupolis, Democritus University of Thrace, 68131 Alexandroupolis, Greece; vasilispetrakis1994@gmail.com; 8Department of Infectious Diseases, HIV Unit, University General Hospital of Alexandroupolis, Democritus University of Thrace, 68131 Alexandroupolis, Greece; 9Department of Orthopaedics and Traumatology, “Venizeleion” General Hospital of Heraklion, 71409 Iraklio, Greece; apopaki@yahoo.gr; 10First Department of Medical Oncology, Metropolitan Hospital of Neon Faliron, 18547 Athens, Greece; samonis@med.uoc.gr; 11School of Medicine, University of Crete, 71003 Heraklion, Greece; 12Department of Orthopedics and Traumatology, Yıldırım Beyazıt University, Ankara City Hospital, 2367 Ankara, Turkey; 132nd Academic Department of Orthopaedics, School of Medicine, National & Kapodistrian University of Athens, 14233 Athens, Greece; di_papadopoulos@yahoo.gr

**Keywords:** coccidioidomycosis, osteoarticular infection, skeletal coccidioidomycosis, antifungal therapy, surgical debridement

## Abstract

Extrapulmonary infections by *Coccidioides* spp., though rare, can occur via dissemination, affecting singular or multiple sites, including the skin and musculoskeletal system. Skeletal involvement often manifests as osteomyelitis, particularly in the axial skeleton. The present systematic review evaluates all documented cases of skeletal coccidioidomycosis to assess the diagnostic and treatment strategies alongside the outcomes, drawing insights from an analysis of 163 verified cases. A systematic review following PRISMA guidelines identified all studies reporting skeletal infections by *Coccidioides* spp. up to 2023 from the PubMed and Scopus databases. Eligible studies evaluated osteoarticular infections from *Coccidioides* spp. Data extraction included demographics, microbiological data, diagnostic methods, and treatment outcomes. Of the 501 initially identified records, a total of 163 patients from 69 studies met the inclusion criteria. Most cases were from the USA, predominantly males, while the median age of the population was 36 years. Diabetes mellitus was the common comorbidity (14.7%). *C. immitis* was the most prevalent pathogen. The spine and hand were common sites of infection (17.5% and 15.1%, respectively). Osteomyelitis by *Coccidioides* spp. was diagnosed, in most cases, by positive cultures (*n* = 68; 41.7%), while, in 49 (30.9%), both the histological examination and cultures yielded the fungus. Surgical debridement was performed in 80.9% of cases. A total of 118 (72.3%) patients were treated with monotherapy, while combination therapy with two or more antifungal agents was reported in 45 (17.7%). Amphotericin B (either liposomal or deoxycholate) was the most commonly given agent as monotherapy in 51 (31.2%) patients, while 30 (18.4%) patients received itraconazole as monotherapy. The rate of infection’s resolution was higher in patients undergoing surgical debridement (79.5%), compared to those treated only with antifungal agents (51.6%, *p* = 0.003). Treatment outcomes showed complete resolution in 74.2% of patients, with a mortality rate of 9.2%. Coccidioidal osseous infections present diagnostic and therapeutic challenges. Surgical intervention is often necessary, complementing antifungal therapy. Vigilance for *Coccidioides* spp. infections, especially in regions with endemicity, is crucial, particularly when bacterial cultures yield negative results.

## 1. Introduction

*Coccidioides*, a dimorphic fungus, may lead to coccidioidomycosis, an uncommon infection primarily impacting the respiratory system [1]. Coccidioidomycosis exhibits remarkable diversity in both its clinical appearance and severity levels. Approximately 50–60% of individuals suffering infections caused by *Coccidioides*, as identified through serological conversion, experience no symptoms. Among those with evident symptomatology, the predominant clinical syndrome manifests as acute respiratory illness, accompanied by pyrexia, cough, and pleuritic pain [2]. Additionally, skin manifestations such as erythema nodosum are frequently apparent in patients suffering from *Coccidioides* infection. In immunocompromised hosts, particularly ones with acquired immunodeficiency syndrome (AIDS), *Coccidioides* infection may cause severe and challenging-to-treat meningitis, while similar manifestations can occasionally occur in immunocompetent individuals. Moreover, infections may result in acute respiratory distress syndrome and potentially fatal pneumonia. The risk of symptomatic infection rises with advancing age [1,2].

The soil fungus *Coccidioides* was discovered in 1892 by a medical student, Alejandro Posadas. This finding occurred in an Argentinian soldier suffering from extensive disease. An analysis of biopsy specimens unveiled organisms bearing a resemblance to the protozoan *Coccidia*, derived from the Greek word “kokkis” meaning “little berry” [2]. *Coccidioides* is a dimorphic fungus, existing in the form of a mycelium in the environment, while, in natural hosts, it is typically found in the form of spherules. Mycelia extend via apical growth, developing genuine septa along their span. These mycelial cells experience autolysis within seven days, resulting in the thinning of their cell walls. Then, the internal septae are developed, forming compartments within the spherules, each harboring endospores [2]. In due course, the endospores containing spherules rupture, liberating those endospores into the surrounding vicinity. Alveolar macrophages subsequently engulf these endospores, initiating a localized host response that culminates in acute inflammation. Endospores have the potential to proliferate within tissues and, upon release into the environment, can promote mycelial growth [1,2].

In susceptible individuals, spherules may occasionally leave the lungs and establish extrapulmonary infections. The most probable paths of dissemination entail the trafficking of macrophages transporting the spherules or endospores. Patients suffering coccidioidomycosis who develop extrapulmonary disease commonly suffer mediastinal lymphadenopathy [2].

Genomic analysis has identified two species of *Coccidioides*: *C. immitis* and *C. posadasii* [2]. However, neither the clinical presentation nor everyday tests allow the distinction between these species [2].

Although rare, extrapulmonary infection can occur through dissemination via lymphatic or hematogenous routes, potentially impacting singular or multiple sites [3]. Approximately 1% of these instances involve the skin, musculoskeletal system, and/or the meninges. Extra-thoracic spreading typically manifests within weeks to months after the initial exposure [4].

In cases where the musculoskeletal system is involved, osteomyelitis is the predominant condition. Additionally, there is a notable tendency for the axial skeleton to be involved [3]. Spinal involvement can range from discitis and soft tissue infection surrounding the vertebrae to vertebral body erosion and nerve compression. When coccidioidomycosis extends beyond the lungs to affect the bones, treatment typically includes appropriate antifungal medication and/or surgical intervention [5]. Usually, initial therapy involves fluconazole or itraconazole, with itraconazole being preferred for bone and joint disease [3].

The current study represents a comprehensive systematic review encompassing all documented reports of osseous coccidioidomycosis affecting both the core and extremities. It aims to evaluate the diagnostic and therapeutic approach, as well as the outcomes based on an analysis of 163 confirmed cases.

## 2. Material and Methods

### 2.1. Search Protocol/Databases

To identify and assess studies that were evaluating the topic of interest of this review, a protocol was designed based on the Preferred Reporting Items for Systematic Reviews and Meta-analyses (PRISMA) guidelines, a methodological protocol for systematic reviews. The review of the literature was conducted between October 2023 and December 2023. The protocol of this review has been registered with Prospero (CRD42023477412).

Two databases, PubMed and Scopus database, were electronically searched until 10 November 2023, for identification of eligible studies with a combination of terms such as “*Coccidioides* spp.”, “infection”, “osseous”, “osteoarticular”, “musculoskeletal”, “bone”, “osteomyelitis”, “septic arthritis”, “orthopedic”, “spondylodiscitis”, and “periprosthetic joint infections”. An initial screen of titles and abstracts of the retrieved articles for eligibility was performed, and studies that were clearly irrelevant to the topic of interest were excluded. The articles that were considered relevant to the topic of this study underwent a thorough full-text evaluation to assess whether they were meeting the inclusion criteria of this review. This process was conducted independently by two authors (LD and SN), while a third investigator (AGT) was called to resolve any disagreement between the first two authors regarding the inclusion of any study. In addition, the reference lists of the articles that underwent full-text evaluation were assessed for identification of any other eligible studies. 

### 2.2. Selection Criteria

Studies were considered eligible if they were evaluating osteoarticular infections from *Coccidioides* spp. Cohort retrospective or prospective studies, observational studies such as case series or case reports, and randomized clinical trials that were published in the English literature were assessed. The inclusion criteria of this review included: (i) a documented infection from *Coccidioides* spp. through a tissue sample biopsy from a osteoarticular location with positive histological findings and/or positive cultures; (ii) a thorough documentation of essential information for each patient including the anatomical location of the infection, any underlying condition, the type of treatment (conservative, surgical, or combination), the antifungal medications (agents, monotherapy, or combination therapy), and the outcome (full resolution or partial resolution). Complete resolution was defined as complete clinical improvement without any clinical signs of infection, while partial resolution was defined as incomplete clinical improvement with partial resolution of the clinical signs of infection, with or without radiological findings of partial resolution of the infection. Studies that were excluded were those that were not involving humans, review studies, studies without documented proof of infection through positive culture or histological findings, and studies without the above-defined essential information for each of the included patients.

### 2.3. Data Extraction and Data Analysis

For the extraction of the reported data from each study, a standard Excel sheet was used by two authors (LD and SN). Information regarding demographics (gender and age) and comorbidities (immunodeficiency, malignancy, intake of corticosteroids, etc.), location of the infection (spine, cranial bones, thoracic ribs, etc.), microbiological data (species, and coexisting infection from another pathogen), and clinical history was recorded. Additionally, data on the method of diagnosis (histology, culture, serology, etc.) and the laboratory findings (white blood cell count, erythrocyte sentiment rate, and C-reactive protein) were extracted. Lastly, information regarding treatment (conservative and/or surgical), type of antifungal therapy (monotherapy or combination therapy), antifungal medications (agents, dose, and duration) and the final outcome (full resolution or partial resolution) was also recorded for each case. 

Statistical analysis included descriptive statistics for the study population. A formal meta-analysis of the extracted data was not feasible since there was high heterogeneity in the collected data; therefore, a limited analysis was performed mainly regarding the association between the final outcome and the type of treatment (surgical and/or conservative, and antifungal monotherapy or combination therapy). The comparison between the evaluated groups was performed using the Chi-square test. The Stata 15.0 software (Stata Corp., College Station, TX, USA) was used for the statistical analysis; statistical significance was set at a *p*-value lower than 0.05.

## 3. Results

### 3.1. Study Characteristics and Patient Demographics

After removing duplicate articles between the two evaluated databases (PubMed and Scopus), 501 records were retrieved (Figure 1). Among them, 320 articles were considered clearly irrelevant to the topic of this systematic review, mainly due to the fact that different pathogens were evaluated, or because these studies were not involving humans and were excluded following a review of their abstracts/titles. Therefore, 320 studies were excluded and the remaining 181 were evaluated thoroughly through a full-text review. Following this full-text review, 92 studies were excluded because the information provided was not considered sufficient based on the inclusion criteria of our review protocol. Hence, 69 studies evaluating 163 patients were finally included as they met the eligibility criteria [5,6,7,8,9,10,11,12,13,14,15,16,17,18,19,20,21,22,23,24,25,26,27,28,29,30,31,32,33,34,35,36,37,38,39,40,41,42,43,44,45,46,47,48,49,50,51,52,53,54,55,56,57,58,59,60,61,62,63,64,65,66,67,68,69,70,71]. Most studies were including patients from the USA, while there were thirteen articles from non-endemic areas (inside and outside the USA) all with a history of travelling for a short or longer period in the endemic area (southern Arizona and California’s southern San Joaquin Valley). Observational comparative studies did not exist and the ones included were case reports or small case series with up to 22 cases. Particularly, among the 68 included studies, there were 51 case reports with just 1 case, 7 reporting 2 cases, 3 reporting 3 cases, 1 reporting 4 cases, 1 reporting 5 cases, 1 reporting 12 cases, 1 reporting 13 cases, and 2 reporting 15 cases, as well as 1 study reporting 22 cases.

The median age of the 163 patients was 36 (Interquartile Range [IQR]: 26–46.5) years, while osteoarticular infections from *Coccidioides* spp. were more commonly seen in men (125 males, 76.6%). The most common underlying disease was diabetes mellitus (DM) (*n* = 24; 14.7%), a well-known predisposing factor for fungal infections, followed by immunodeficiency due to a disease or immunosuppressive therapy (*n* = 12; 7.3%), and sarcoidosis (*n* = 8; 4.8%). Notably, 37 (22.6%) patients were on corticosteroids. The reason and duration of the corticosteroid therapy was not mentioned in most studies. Moreover, in 14 patients (8.5%), a previous fungal infection before the onset of the symptoms from the coccidioidal infection was reported. The causative pathogens were *Candida* spp. in 7 of them, and *Aspergillus* spp. in 1 of them, while in the rest of these patients the causative pathogen was not mentioned. The underlying conditions and demographics of the study population are summarized in Table 1.

### 3.2. Microbiology and Location

In most cases, the causative pathogen was *C. immitis* (*n* = 103; 75.7%), while, in two cases (*n* = 2; 1.2%), *C. possadasii* was isolated, and, in 35 cases (*n* = 58; 35.5%), the causative pathogen was not specified (Table 2).

Regarding the anatomical distribution of these infections, 165 infected foci were identified in the 163 included patients, with two different anatomic locations being evident in 2 (1.2%) patients. The most common location for osteoarticular infections from *Coccidioides* spp. was the spine with 32 (19.3%) cases of spondylodiscitis due to *Coccidioides* spp. The osseous structures of the knee were another common location with 29 (17.5%) cases reported in this region (Table 3). Other common locations for osteoarticular infections from *Coccidioides* spp. included the osseous structures of the hand (*n* = 25; 15.1%), while the pelvic bones were involved in 14 (8.4%) cases.

### 3.3. Diagnostic Studies and Markers of Inflammation

Osteomyelitis by *Coccidioides* spp. was diagnosed by positive cultures in 68 (41.7%) patients, while, in 49 (30.9%), *Coccidioides* spp. were identified by both histological examination (through direct microscopy) and cultures. In 37 (22.6%) patients, the diagnosis of osteomyelitis by *Coccidioides* spp. was confirmed only by positive histological findings, in 3 (1.8%), by positive serology and cultures, and, in 3 (1.8%), the diagnosis was established by positive histological findings and serology. Moreover, in 3 (1.8%) patients, these infections were diagnosed by positive cultures, histology examination, and positive serological findings (Table 4).

Regarding the laboratory findings, the median white blood cell count (WBC) was 10.100 (IQR: 6.900–13.500)/mL, while the median C-reactive protein (CRP) was 19 (4–94) mg/L and the median erythrocyte sedimentation rate (ESR) was 61 (27–102) mm/hr. Interestingly, only 54 (33.1%) patients had an elevated (>12.000/mL) WBC count and only 91 (55.8%) had elevated CRP levels, as opposed to the ESR levels which were abnormal in a higher percentage of patients (137 patients; 73.6%, Table 4)

### 3.4. Antibiotic Agents, Protocols, and Treatment Outcomes

Overall, surgical debridement was performed in 132 (80.9%) patients, while conservative treatment with antifungal therapy was followed in 31 (19.1%) (Table 5). Regarding the antifungal medications, monotherapy with one antifungal agent was provided in 118 (72.3%) patients, while combination/sequential therapy with two or more antifungal agents was given in 45 (17.7%) patients. In 37 of these 45 patients, the antifungal regime included some period of concomitant use of antifungal agents, while, in the remaining 8 patients, the antifungal regime included a sequential therapy in which one antifungal agent was given and then replaced by another one. Amphotericin B (either liposomal or deoxycholate) was given as monotherapy in 51 (31.2%) patients for a median duration of 8 (IQR: 6–13) months, while 30 (18.4%) patients received itraconazole as monotherapy for a median period of 8 (IQR: 6–13) months, and 28 (17.1%) received fluconazole as monotherapy (Table 5). Moreover, a combination/sequential therapy with amphotericin B and itraconazole was given in 15 (9.2%) patients, a combination/sequential therapy with amphotericin B and ketoconazole in 13 (7.8%), while a combination/sequential therapy with amphotericin B and voriconazole was given in 5 (3.0%) (Table 5). Overall, amphotericin B was administered in 91 (55.5%) patients. The median total dose of amphotericin B was 2000 (750–4000) mg, while the median daily dose of itraconazole was 400 (400–400) mg.

Regarding the outcome, complete resolution of the infection was achieved in 121 (74.2%) patients, while partial resolution was reported in 42 (15.8%). The overall mortality rate related to the infection was 9.2% (*n* = 15). The rate of complete resolution of the infection was higher in patients in whom surgical debridement was performed (*n* = 105 out of 132 patients; 79.5%), compared to those in whom conservative treatment with antifungal medications was followed (*n* = 16 out of 31 patients; 51.6%, *p* = 0.003). Conversely, the rate of complete infection resolution was similar in patients who received antifungal monotherapy and those who received combination/sequential antifungal therapy (*n* = 89 out of 118 patients; 75.4% vs. *n* = 32 out of 45 patients; 71.1%; *p* = 0.55). Lastly, the rate of complete infection resolution was also similar between patients who received amphotericin B as monotherapy (*n* = 40; 78.4%) and patients who received itraconazole as monotherapy (*n* = 22; 73.3%; *p* = 0.60).

## 4. Discussion

Coccidioidomycosis, commonly referred to as “valley fever” or “desert rheumatism”, represents a fungal infection triggered by the soil-dwelling fungi *C. immitis* and *C. posadasii* most commonly found in several endemic regions [3,72]. Inhalation typically initiates infection, often without noticeable symptoms. Nevertheless, the disease may also manifest as pneumonia, ranging from mild to severe, and may escalate to life-threatening conditions with dissemination to other body sites, including the skin, bones, joints, and meninges [73]. Around 70% of coccidioidal infections in the USA are detected in Arizona, with an additional 25% documented in California. In the southwestern US region alone, roughly 150,000 new cases are reported every year, indicating a consistent increase in the incidence of coccidioidomycosis [72,74,75]. This systematic review aimed to investigate the epidemiology, patients’ characteristics, symptoms, and treatment options, as well as outcomes, of osseous infections caused by *Coccidioides* spp.

The musculoskeletal system is frequently affected in disseminated coccidioidomycosis, with osseous involvement occurring in approximately 10 to 50% of cases [3,75]. The present study showed that the most common location for osteoarticular infections from *Coccidioides* spp. was the spine with 32 (19.3%) cases of spondylodiscitis. The osseous structures of the knee were another common location with 29 (17.3%) cases reported in this anatomical region. Multiple-site infections were also reported (two different anatomic locations in 2 (1.2%) patients). The spine is a common site for infections, due to its unique vascular supply and its location in the human body. Therefore, hematogenous spread from distant infected foci can occur, allowing the seeding of the infection in the spine, while contiguous spread from adjacent organs in the retroperitoneal space or the thoracic cavity can also occur.

Two species of *Coccidioides*, namely, *C. immitis* and *C. posadasii*, are accountable for infections in humans. In this systematic review, *C. immitis* was identified in 103 cases (75.7%), while, in 2 cases (*n* = 2; 1.2%), *C. possadasii* was isolated. Moreover, in 35 cases (*n* = 58; 35.5%), the causative pathogen was not further characterized. Although fungal culture enables the identification of genus *Coccidioides*, distinguishing *C. immitis* from *C. posadasii* requires PCR and genomic analysis, which are not widely accessible in routine clinical settings. Despite genomic disparities between the two species, no discernible differences in disease manifestation, diagnosis, or treatment have been noted. Indeed, their pathogenicity appears to be largely comparable [76]. In all examined cases, the causative agents were identified through histopathology, cultures, and/or serology. Particularly, diagnoses were possible by cultures in 68 (41.7%) patients, followed by histological findings, in 37 (22.6%). Positive cultures may serve as the earliest and, sometimes, the sole diagnostic method. However, the sensitivity of cultures for the identification of *Coccidioides* spp. is not high; therefore, in the case of negative results, a further diagnostic evaluation is needed if such pathogens are suspected [3]. Genetic probing detecting *Coccidioides*-specific DNA is rapid and reliable, although it may not differentiate between *C. immitis* and *C. posadasii*. Antigen detection methods in urine or blood samples are available for cases with disseminated infection. Real-time polymerase chain reaction (RT-PCR) testing, while highly specific, may have a lower sensitivity compared to cultures but offers faster results, especially for formalin-fixed tissues, and is increasingly available in specialized laboratories [3,77].

Antifungal treatment is highly recommended in individuals suffering from osseous and joint coccidioidal infection [78]. Triazole agents, alongside amphotericin B, represent the commonly prescribed antifungal medications for managing coccidioidomycosis [73]. The results of the current study revealed that 118 patients (72.3%) were managed with a single antifungal regimen, while 45 (17.7%) underwent combination therapy involving two or more antifungal agents. Monotherapy with either liposomal or deoxycholate Amphotericin B was prescribed to 51 patients (31.2%) for a median duration of 8 months, while 30 patients (18.4%) received Itraconazole alone for a median period of 8 months, and 28 (17.1%) were treated solely with Fluconazole. Additionally, combination or sequential therapy involving Amphotericin B and Itraconazole was administered to 15 patients (9.2%), Amphotericin B and Ketoconazole to 13 (7.8%), and Amphotericin B and Voriconazole to 5 (3.0%). Overall, Amphotericin B was utilized in 91 patients (55.5%). The median total dose of Amphotericin B was 2000 mg (750–4000), while the median daily dose of Itraconazole was 400 mg (400–400). Conversely, the rate of complete infection resolution was similar between patients treated with antifungal monotherapy and those receiving combination/sequential antifungal therapy (75.4% vs. 71.1%; *p* = 0.55). Similarly, there was no significant difference in the rate of complete infection resolution between patients treated with Amphotericin B as monotherapy (78.4%) and those treated with itraconazole as monotherapy (73.3%; *p* = 0.60).

The relative toxicities of Amphotericin B and associated side effects, notably renal dysfunction, limit its prolonged use [79,80,81]. Although the nephrotoxicity of amphotericin B has been reduced by the development of liposomal formulations, the prolonged administration of this agent may still negatively impact kidney function [82]. Fluconazole is acknowledged as the most commonly used antifungal medication for treating coccidioidomycosis, primarily due to its affordability and availability in both intravenous and oral formulations [83]. Fluconazole is notably distinguished by its excellent oral bioavailability, which remains unaffected by food or gastrointestinal conditions. Furthermore, it is characterized by minimal protein binding, facilitating widespread distribution across various human tissues and fluids [3]. Additionally, Itraconazole is extensively utilized, with some evidence suggesting its superiority in treating certain disseminated coccidioidal infections [3]. In cases of severe osteoarticular disease, such as limb-threatening skeletal conditions or vertebral infections posing an imminent risk of spinal cord compromise, combination therapy involving a triazole agent and amphotericin B is recommended. There are some reports in the literature that combination therapy can be used in severe cases of these fungal infections, although there is not any evidence for improved outcomes in osteoarticular disease [78].

Surgical intervention may be necessary depending on the severity of the osseous coccidioidal infections [84,85]. In total, surgical debridement was carried out in 132 patients (80.9%), while 31 (19.1%) were treated conservatively with antifungal therapy. This review revealed that complete resolution of the infection was achieved in 121 patients (74.2%), while partial resolution was reported in 42 (15.8%). The overall mortality rate related to the infection was 9.2% (*n* = 15). Patients undergoing surgical debridement had a significantly higher rate of complete infection resolution (105 out of 132 patients; 79.5%) compared to those who were treated conservatively (16 out of 31 patients; 51.6%; *p* = 0.003). Vertebral instability, neurological deterioration/impairment, and infection progression despite antifungal therapy initiation should be carefully assessed as potential indicators for surgical intervention [3,82].

The current review possesses certain limitations. The most significant limitation is that most cases lack descriptions of antifungal agent dosages, drug serum levels, minimum inhibitory concentrations, and associated side effects. Another significant limitation is that a formal meta-analysis of the extracted data was not feasible, since there was high heterogeneity in the collected data; therefore, a limited analysis was performed mainly regarding the association between the final outcome and the type of treatment. It must be noted that, a couple of years ago, our team had published a narrative review regarding skeletal infections by *Coccidioides* spp. [3]. Nevertheless, the present one is a systematic review, including all reported cases without time limits, offering, therefore, much more valuable information regarding the diagnostic and therapeutic approach, as well as the outcome of osseous coccidioidal infections. Lastly, it would be valuable to have more information regarding patients in whom combination therapy (concomitant use of antifungal agents) was given vs. those in whom sequential therapy (one antifungal agent was given, and then replaced by another one) was provided. Unfortunately, separate data regarding these two groups were not available; therefore, a further analysis and comparison between these two groups could not be performed.

## 5. Conclusions

Osseous infections caused by *Coccidioides* spp. pose a significant medical challenge, necessitating timely and precise diagnosis alongside multidisciplinary management. Surgical intervention is often imperative in most instances. The combination of sustained and appropriate antifungal therapy coupled with surgical intervention appears to represent the most effective therapeutic strategy presently available. Additionally, in cases of osseous infections, particularly in the southwestern United States when cultures yield negative results for bacteria and/or cocci, a heightened level of suspicion for *Coccidioides* spp. is warranted.

## Figures and Tables

**Figure 1 jof-10-00270-f001:**
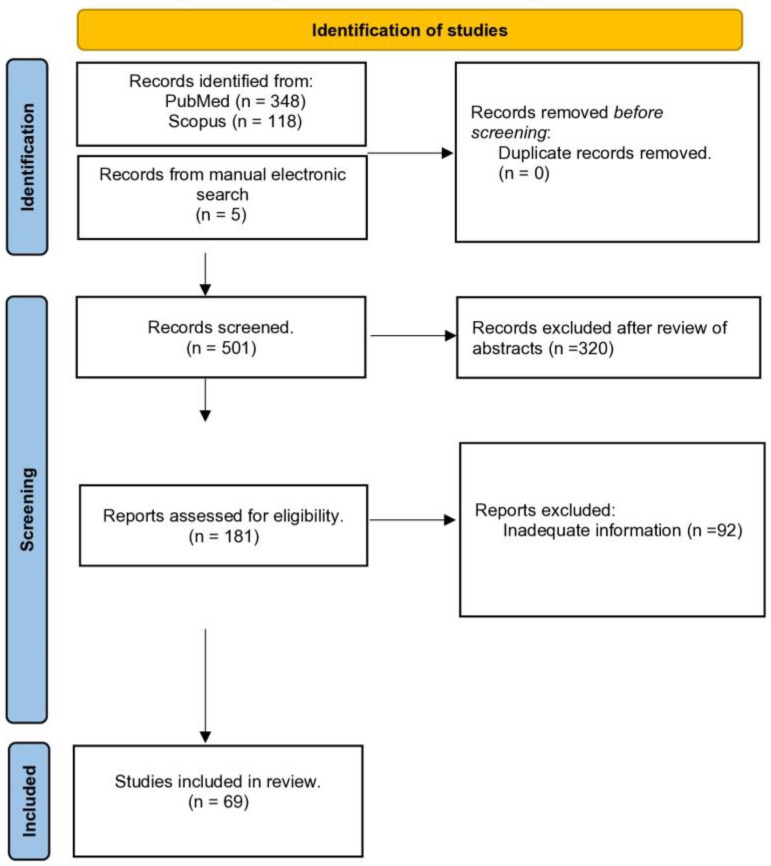
PRISMA 2020 flow diagram for the methodology of this systematic review of studies evaluating osteoarticular infections from *Coccidioides* spp. until 10 November 2023.

**Table 1 jof-10-00270-t001:** Demographics of the study population.

Variables	Patients (*n* = 163)
Age (years)	37.7 ± 18.0, 36 (26–46.5)
Gender (male)	125 (76.6)
Comorbidities/Underlying condition	
Previous fungal infection	14 (8.5)
Hematologic neoplasia	8 (4.9)
Immunodeficiency/immunosuppression therapy	12 (7.3)
Diabetes	24 (14.7)
Tuberculosis	4 (2.4)
Sarcoidosis	8 (4.8)
None	14 (8.5)
Not mentioned	79 (48.4)
Corticosteroids	37 (22.6)

Footnote: Data are presented as means ± SD, medians and interquartile ranges (IQR), or as absolute frequencies (percentages) when appropriate.

**Table 2 jof-10-00270-t002:** Microbiology of *Coccidioides* species and co-cultured bacterial pathogens.

Genus	Pathogen	Patients (*n* = 163)
*Coccidioides* spp.	*C. immitis*	103 (75.7)
	*C. posadasii*	2 (1.2)
	Not specified	58 (35.5)
Co-cultured bacterial pathogens	*Staphylococcus* spp.	6 (3.6)
	*Enterobacterales* spp.	3 (1.8)
	*Pseudomonas aeruginosa*	1 (0.6)

Footnote: Data are presented as absolute frequencies (percentages).

**Table 3 jof-10-00270-t003:** Anatomical distribution infections.

Location	Infected Foci (*n* = 165)
Spine	32 (19.3)
Ribs and sternum	6 (3.6)
Lower extremities	
Knee	29 (17.5)
Foot	15 (9.0)
Ankle	4 (2.4)
Other areas	13 (7.8)
Cranial bones	10 (6.0)
Upper extremity	
Wrist	9 (5.4)
Hand	25 (15.1)
Elbow	6 (3.6)
Shoulder	2 (1.2)
Pelvis (iliac, sacral)	14 (8.4)

Footnote: Data are presented as absolute frequencies (percentages).

**Table 4 jof-10-00270-t004:** Diagnostic studies and markers of inflammation.

	Patients (*n* = 163)
Direct culture	68 (41.7)
Histology and direct culture	49 (30.0)
Histology	37 (22.6)
Direct cultures and serology	3 (1.8)
Histology and serology	3 (1.8)
Direct cultures, histology and PCR	3 (1.8)
WBC count (×10^3^/mL)	11.0 ± 6.1, 10.1 (6.9–13.5)
Abnormal WBC count (>12 × 10^3^/mL)	54 (33.1)
ESR (mm/h)	65.4 ± 42.3, 61.0 (27.0–102.0)
Abnormal ESR (>20 mm/h)	137 (73.6)
CRP (mg/L)	46.1 ± 56.0, 19.0 (4.0–94.0)
Abnormal CRP (>10 mg/L)	91 (55.8)

Footnote: Data are presented as means ± SD, medians and interquartile ranges (IQR), or as absolute frequencies (percentages) when appropriate. Abbreviations: PCR, polymerase chain reaction; WBC, white blood cell; ESR, erythrocyte sedimentation rate; CRP, C-reactive protein.

**Table 5 jof-10-00270-t005:** Treatment protocols and rates of complete infection resolution.

	Patients (*n* = 163)	Duration (Months)	Complete Resolution
Total	163 (100.0)	8.8 ± 4.3, 8 (6–13)	121 (74.2)
Amphotericin B monotherapy	51 (31.2)	8.7 ± 4.7, 8 (6–13)	40 (78.4)
Fluconazole monotherapy	28 (17.1)	19.2 ± 5.1, 18 (11–29)	20 (71.4)
Itraconazole monotherapy	30 (18.4)	9.7 ± 16.7, 6 (6–9)	22 (73.3)
Ketoconazole monotherapy	4 (2.4)	-	3 (75.0)
Voriconazole monotherapy	5 (3.0)	-	4 (80.0)
Amphotericin B + itraconazole	15 (9.2)	-	10 (66.6)
Amphotericin B + ketoconazole	13 (7.8)	-	9 (69.2)
Amphotericin B + fluconazole	4 (2.4)	-	3 (75.0)
Amphotericin B + voriconazole	5 (3.0)	-	4 (80.0)
Other	8 (4.9)	-	6 (75.0)
Surgical debridement	132 (80.9)	-	105 (79.5)
Antifungal monotherapy	118 (72.3)	8.8 ± 4.3, 8 (6–13)	89 (75.4)

Footnote: Data are presented as means ± SD, medians and interquartile ranges (IQR), or as absolute frequencies (percentages) when appropriate.

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
