# Peer review of "Diagnosis, Treatment, and Outcome of Coccidioidal Osseous Infections: A Systematic Review"

_jof, 2024, doi:10.3390/jof10040270_

Round 1
Reviewer 1 Report
Comments and Suggestions for Authors
Dear editors,
The authors presented a systematic review of case reports and case series, including 163 patients with coccidioidomycosis affecting the osteoarticular system. This manifestation of systemic mycosis, which normally affects the lungs and central nervous system, is relatively uncommon. Consequently, these findings have significant value for physicians and surgeons involved in the management of cases of osteoarticular coccidioidomycosis.
However, it is important to recognize that although the study offers valuable information, the topic itself is not new and the results do not substantially contribute to advancing current understanding of the disease. The same authors previously published a narrative review on this topic (Koutserimpas C, Naoum S, Raptis K, Vrioni G, Samonis G, Alpantaki K. Skeletal Infections Caused by Coccidioides Species. Diagnostics (Basel). 2022;12(3):714. doi: 10.3390/diagnostics12030714), and more recently another review appeared in the Journal of Fungi (Moni BM, Wise BL, Loots GG, Weilhammer DR. Coccidioidomycosis Osteoarticular Dissemination. J Fungi (Basel). 2023;9(10):1002. Published 2023 Oct 11. doi:10.3390/jof9101002). Despite this, I still recommend publishing this article, as long as the authors implement the following adjustments:
Abstract:
Lines 41-42- “Most cases were from the USA, predominantly males with median age of 36 years."
Note that this phrase may suggest that 36 years is the median age specifically for males.
Line 44- “Elevated inflammatory markers were inconsistent”.
This statement is meaningless.
Line 45- “Osteomyelitis by Coccidioides spp. was diagnosed in most cases by positive cultures (n=68;41.7%) patients”
The word “patients” is unnecessary.
Introduction
Line 78- “Coccidioides species commonly appear as mycelia in both natural hosts and laboratory settings”
Coccidioides spp. is a dimorphic fungus, existing in the form of mycelium in the environment, while in natural hosts, it is typically found in the form of spherules.
Material and methods
Line 143. “Studies that were excluded were those that ……and studies with insufficient information for the included patients.”
Please, note that this exclusion criterion (insufficient information) is imprecise. To ensure transparency, it is necessary to describe what information was considered sufficient. For example, by what criteria were the following articles excluded? They apparently contain all the information shown in the present study.
- Ho AK, Shrader MW, Falk MN, Segal LS. Diagnosis and initial management of musculoskeletal coccidioidomycosis in children. J Pediatr Orthop. 2014;34(5):571-577. doi:10.1097/BPO.0000000000000147. All patients (n=20) had positive cultures or a diagnostic histologic evaluation of surgical biopsy or debridement samples.
- Ricciotti RW, Shekhel TA, Blair JE, Colby TV, Sobonya RE, Larsen BT. Surgical pathology of skeletal coccidioidomycosis: a clinical and histopathologic analysis of 25 cases. Am J Surg Pathol. 2014;38(12):1672-1680. doi:10.1097/PAS.0000000000000284
Lines 141, 155. Please define in the methods section what constitutes “complete and partial resolution”. Does it refer to the disappearance of symptoms and signs? Normalization of inflammatory markers? Restoration of bone on radiological images?
Lines 151. What is meant by "direct culture"? Does it refer to culture without prior decontamination or enrichment? Please provide clarification on this point.
Results
Why was direct microscopy not considered or discussed as a diagnostic method?
Lines 238-240, 253, 369, 373, 380, 396. Please clarify what constitutes cases treated with combination/sequential therapy. Are they a combination of patients treated with more than one antifungal, including those who used them concurrently and those who initially used one antifungal followed by another? Given that there is no formal recommendation for the concomitant use of multiple antifungals in coccidioidomycosis, it is essential for the authors to distinguish and analyse separately the outcomes of cases using concomitant therapy from those using sequential therapy. This differentiation should also be addressed in the discussion.
Discussion
The authors should revise the discussion, as it is inappropriate to restate the results within it. (E.g: lines 284-329)
Lines 301-307: “The present study showed that the most common location for osteoarticular infections from Coccidioides spp. was the spine with 29 (17.5%) cases of spondylodiscitis. The osseous structures of the hand were another common location with 25 (15.1%) cases reported in this anatomical region. Other common locations included the osseous structures of the foot (n=15; 9.0%) and the pelvis (n=14; 8.4%). Multiple-site infections were also reported (2 different anatomic locations in 2 (1.2%) patients).”
These are merely results, lacking any commentary on the underlying reasons for these findings.
Lines 311. “Although fungal culture enables the identification of Coccidioides species, distinguishing C. immitis requires PCR and genomic analysis, which are not widely accessible in routine clinical settings.
Please, replace this sentence with: Although fungal culture enables the identification of genus Coccidioides, distinguishing C. immitis from C. posadasii requires PCR and genomic analysis, which are not widely accessible in routine clinical settings.
Line 323. “In all examined cases, the causative species were identified through histopathology, cultures, and/or serology”.
Histopathological examination, cultures and/or serology enable the identification of the species of Coccidioides. Please adjust this sentence, replacing “species” with “agent”.
Lines 341. “Positive cultures may serve as the earliest and sometimes the sole diagnostic method, but additional testing is often necessary to confirm Coccidioides spp”
This statement may lead the reader to believe that a positive culture requires confirmation by another method, which is not true. The authors may have intended to convey that in the event of a negative culture, alternative diagnostic methods will be necessary, as the sensitivity of culture alone may be low. Please provide clarification on this point.
Lines 395-399. “In cases of severe osteoarticular disease, such as limb-threatening skeletal conditions or vertebral infections posing an imminent risk of spinal cord compromise, combination therapy involving a triazole agent and amphotericin B is recommended”.
Did the authors mean to say that treatment with more than one antifungal drug simultaneously has been recommended for severe cases of osteoarticular coccidioidomycosis? Could you please provide a reference? The two references cited after the following sentence do not support this statement.
There are typing and formatting errors that need to be adjusted. There may possibly be a request from the journal pointing out the necessary adjustments. Here are some of them:
Do not capitalize the names of diseases:
Line 61: …Coccidioidomycosis…Replace with coccidioidomycosis
Line 198 and table 2. C. immitis instead of C. Immitis
Line 199 and table 2. C. posadasii instead of C. possadasii
Lines 50, 233-243, 255. Do not capitalize the generic names of drugs. (itraconazole, amphotericin B, fluconazole, voriconazole)
Line 244. Table 5. Correct: ketoconzole, fluzonazole
Line 202. Enterobacterales instead of Enterobacterales. Order and suborder are not italicized.
Line 204 Table 2. Staphylococcus instead of Staphylococcus
Table 2. Pseudomonas aeruginosa instead of Pseudomonas aeruginosa
Italicize family, genus, species, and variety or subspecies
Lines 286. The plural of fungus is “fungi”.
Author Response
Dear Editor,
We would like to thank you and the reviewers for evaluating our manuscript entitled: “Diagnosis, treatment and outcome of Coccidioidal osseous infections: A systematic review”. In the revised version of the manuscript, we tried to address adequately all points raised by the reviewers. Our responses to each point brought up are provided below. (Original comments in bold). Changes in the revised manuscript have been highlighted in red.
Reviewer 1
Abstract:
Lines 41-42- “Most cases were from the USA, predominantly males with median age of 36 years." Note that this phrase may suggest that 36 years is the median age specifically for males.
Authors’ response: This phrase has been modified to clear out that 36 years is the median age for the whole population and not only for males.
Line 44- “Elevated inflammatory markers were inconsistent”. This statement is meaningless.
Authors’ response: This phrase has been deleted.
Line 45 “Osteomyelitis by Coccidioides spp. was diagnosed in most cases by positive cultures (n=68;41.7%) patients”. The word “patients” is unnecessary.
Authors’ response: The word “patients” has been deleted.
Introduction
Line 78- “Coccidioides species commonly appear as mycelia in both natural hosts and laboratory settings”. Coccidioides spp. is a dimorphic fungus, existing in the form of mycelium in the environment, while in natural hosts, it is typically found in the form of spherules.
Authors’ response: We thank the reviewer for making this valid comment. This information is now provided and the previous phrase has been modified in the revised manuscript.
Material and methods:
Line 143. “Studies that were excluded were those that ……and studies with insufficient information for the included patients.” Please, note that this exclusion criterion (insufficient information) is imprecise. To ensure transparency, it is necessary to describe what information was considered sufficient. For example, by what criteria were the following articles excluded? They apparently contain all the information shown in the present study.
- Ho AK, Shrader MW, Falk MN, Segal LS. Diagnosis and initial management of musculoskeletal coccidioidomycosis in children. J Pediatr Orthop. 2014;34(5):571-577. doi:10.1097/BPO.0000000000000147. All patients (n=20) had positive cultures or a diagnostic histologic evaluation of surgical biopsy or debridement samples.
- Ricciotti RW, Shekhel TA, Blair JE, Colby TV, Sobonya RE, Larsen BT. Surgical pathology of skeletal coccidioidomycosis: a clinical and histopathologic analysis of 25 cases. Am J Surg Pathol. 2014;38(12):1672-1680. doi:10.1097/PAS.0000000000000284
Authors’ response: We agree with the reviewer that the exclusion criteria have to be specified in order to ensure transoarency. Studies were excluded if they were not containing any essential information for each of the included patients. As mentioned in the “Selection criteria” paragraph of the manuscript, essential information included the anatomical location of the infection, the type of treatment (conservative, surgical, combination), the antifungal therapy (agents, monotherapy vs. combination therapy), and the outcome (complete resolution [defined as complete clinical improvement without any clinical signs of infection], partial resolution [defined as incomplete clinical improvement with partial resolution of the clinical signs of infection, with or without radiological findings of partial resolution of the infection]. This is now highlighted in the revised manuscript.
Regarding the study by Ho et al, although there is information about treatment in all 20 patients (fluconazole was used in 15 patients, itraconazole in 2 patients, voriconazole in 2 patients, and amphotericin B in 1 patient, while surgical debridement was utilized in 10 patients), there is no information regarding the outcome (full vs partial resolution).
Regarding the study by Ricciotti et al, although we agree with the reviewer that this study could be included in our review, it was decided to be excluded because there was missing data regarding medical treatment (type of antifungal agents) in some of the included patients (in 1 out of the 25 patients). Moreover, although it was mentioned that recurrence occurred in 10 out of the 25 included patients, it was not mentioned if complete or partial resolution occurred in the rest of the patients.
Lines 141, 155. Please define in the methods section what constitutes “complete and partial resolution”. Does it refer to the disappearance of symptoms and signs? Normalization of inflammatory markers? Restoration of bone on radiological images?
Authors’ response: We agree with the reviewer that this is valuable information to the readers. Complete resolution was defined as complete clinical improvement without any clinical signs of infection, and partial resolution was defined as incomplete clinical improvement with partial resolution of the clinical signs of infection, with or without radiological findings of partial resolution of the infection. Complete and partial resolution are now defined in the revised manuscript.
Lines 151. What is meant by "direct culture"? Does it refer to culture without prior decontamination or enrichment? Please provide clarification on this point.
Authors’ response: The word “direct” has been deleted since it does not add any meaning.
Results
Why was direct microscopy not considered or discussed as a diagnostic method?
Authors’ response: Histological examination is mentioned as one of the diagnostic methods for our included cases. Positive histology findings refer to positive findings by direct microscopy as well. This is now clarified in the revised manuscript.
Lines 238-240, 253, 369, 373, 380, 396. Please clarify what constitutes cases treated with combination/sequential therapy. Are they a combination of patients treated with more than one antifungal, including those who used them concurrently and those who initially used one antifungal followed by another? Given that there is no formal recommendation for the concomitant use of multiple antifungals in coccidioidomycosis, it is essential for the authors to distinguish and analyse separately the outcomes of cases using concomitant therapy from those using sequential therapy. This differentiation should also be addressed in the discussion.
Authors’ response: We agree with the reviewer that it would be valuable to have more information regarding patients in whom combination therapy (concomitant use of antifungal agents) was given vs those in whom sequential therapy (one antifungal agent was given and then replaced by another one) was provided. Overall, combination/sequential therapy was given in 45 patients. In 37 of them, the antifungal regime included some period of concomitant use of antifungal agents, while in the rest 8 the antifungal regime included a sequential therapy in which one antifungal agent was given and then replaced by another one. Although this information is now given in the revised manuscript, unfortunately separate data regarding these two groups were not available, therefore further analysis and comparison between these two groups cannot be performed. This poses a certain limitation of the study and it is now discussed in the limitation paragraph of the manuscript.
Discussion
The authors should revise the discussion, as it is inappropriate to restate the results within it. (E.g: lines 284-329)
Authors’ response: Based on the reviewer’s recommendation, the Discussion has been revised and certain parts of the manuscript that were restating the results have been deleted, while others have been modified.
Lines 301-307: “The present study showed that the most common location for osteoarticular infections from Coccidioides spp. was the spine with 29 (17.5%) cases of spondylodiscitis. The osseous structures of the hand were another common location with 25 (15.1%) cases reported in this anatomical region. Other common locations included the osseous structures of the foot (n=15; 9.0%) and the pelvis (n=14; 8.4%). Multiple-site infections were also reported (2 different anatomic locations in 2 (1.2%) patients).”
These are merely results, lacking any commentary on the underlying reasons for these findings.
Authors’ response: We agree with the reviewer that some discussion regarding the underlying reasons for our findings about the location of the infections is needed. In the revised manuscript a short discussion regarding reasons for the spine being the most common location for coccidioidal infections has been added.
Lines 311. “Although fungal culture enables the identification of Coccidioides species, distinguishing C. immitis requires PCR and genomic analysis, which are not widely accessible in routine clinical settings.
Please, replace this sentence with: Although fungal culture enables the identification of genus Coccidioides, distinguishing C. immitis from C. posadasii requires PCR and genomic analysis, which are not widely accessible in routine clinical settings.
Authors’ response: We thank the reviewer for his/ger suggestion. This phrase has been replaced by the one proposed by the reviewer.
Line 323. “In all examined cases, the causative species were identified through histopathology, cultures, and/or serology”.
Histopathological examination, cultures and/or serology enable the identification of the species of Coccidioides. Please adjust this sentence, replacing “species” with “agent”.
Authors’ response: The word “species” has been replaced by “agents” in the revised manuscript.
Lines 341. “Positive cultures may serve as the earliest and sometimes the sole diagnostic method, but additional testing is often necessary to confirm Coccidioides spp”
This statement may lead the reader to believe that a positive culture requires confirmation by another method, which is not true. The authors may have intended to convey that in the event of a negative culture, alternative diagnostic methods will be necessary, as the sensitivity of culture alone may be low. Please provide clarification on this point.
Authors’ response: We agree with the reviewer that this phrase needs to be restated. Indeed, we meant to say that in case of negative results, further diagnostic tests are needed if strong clinical suspicion exists. This phrase has been modified accordingly in the revised manuscript.
Lines 395-399. “In cases of severe osteoarticular disease, such as limb-threatening skeletal conditions or vertebral infections posing an imminent risk of spinal cord compromise, combination therapy involving a triazole agent and amphotericin B is recommended”.
Did the authors mean to say that treatment with more than one antifungal drug simultaneously has been recommended for severe cases of osteoarticular coccidioidomycosis? Could you please provide a reference? The two references cited after the following sentence do not support this statement.
Authors’ response: We agree with the reviewer that this needs further clarification. Although combination therapy has not been included in any official guidelines for the treatment of musculoskeletal infections from Coccidioides spp., there are reports in the literature that combination therapy can be used for severe infections. For example, in the recommendations by Galgiani et al it is mentioned that “Some physicians initiate therapy with intrathecal amphotericin B in addition to an azole on the basis of their belief that responses are more prompt with this approach” in cases of meningitis [1]. However, we have to highlight that these reports do not refer to osteoarticular infections. The phrase in the manuscript has been revised to point out that there are only some reports in the literature that combination therapy can be used in case of severe coccidioidal infections, without any evidence for improved outcomes in osteoarticular disease.
- John N. Galgiani, Neil M. Ampel, Antonino Catanzaro, Royce H. Johnson, David A. Stevens, Paul L. Williams, Practice Guidelines for the Treatment of Coccidioidomycosis, Clinical Infectious Diseases, Volume 30, Issue 4, April 2000, Pages 658–661
There are typing and formatting errors that need to be adjusted. There may possibly be a request from the journal pointing out the necessary adjustments. Here are some of them:
Do not capitalize the names of diseases: Line 61: …Coccidioidomycosis…Replace with coccidioidomycosis
Line 198 and table 2. C. immitis instead of C. Immitis
Line 199 and table 2. C. posadasii instead of C. possadasii
Lines 50, 233-243, 255. Do not capitalize the generic names of drugs. (itraconazole, amphotericin B, fluconazole, voriconazole)
Line 244. Table 5. Correct: ketoconzole, fluzonazole
Line 202. Enterobacterales instead of Enterobacterales. Order and suborder are not italicized.
Line 204 Table 2. Staphylococcus instead of Staphylococcus
Table 2. Pseudomonas aeruginosa instead of Pseudomonas aeruginosa
Italicize family, genus, species, and variety or subspecies
Lines 286. The plural of fungus is “fungi”.
Authors’ response: We thank the reviewer for pointing out these errors. These errors have been corrected throughout the manuscript.
On behalf of all the authors,
Yours sincerely,
Andreas G. Tsantes, MD
Laboratory of Haematology and Blood Bank Unit, "Attiko" Hospital, School of
Medicine, National and Kapodistrian University of Athens, Athens, Greece
Reviewer 2 Report
Comments and Suggestions for Authors
I have reviewed the manuscript entitled “Diagnosis. Treatment and outcome of Coccidiodal osseous infections: A systematic review” submitted by Tsantes, AG, et al. The review describes findings from a broad PubMed and Scopus databases evaluating involvement of osteoarticular structures with Coccidioides spp. Overall, the investigators identified 163 patients that met the inclusion criteria. The manuscript I think would be of interest to the readers of the journal. It is relatively well written and organized and relatively easy to understand. However, there are a few items that should be addressed prior to accepting for publication.
Comments:
Materials and Methods:
Line 124-25: confusing, need to re-structure.
Line 128-130: Confusing, should be re-structured.
Results:
Line 176: change to “patients” not “patents”.
Line: authors state that the most common underlying condition was “ diabetes”. However, the most common condition per the Table 1 was steroids. This should be corrected. In addition to the “immunodeficiency state” which in text is 7.3%, however steroid use, which is an immunodeficiency per table is 22.6 %. Please address and correct.
Line 193: what does a previous fungal infection” mean. Prior cocci vs others. If so what type of fungal infection?
Line 200-203: Not sure what the co-bacterial infections mean. Maybe just colonization fro biopsy. Since it is very low rate, I would just delete.
Line 207-208: According to the table, the most common site of infection was the knee at 19.3%, not the spine. Please correct.
Line 231-232:”followed” is not appropriate here. Consider change to “provided”.
Table 6 does not provide useful data. Consider deleting.
Line 246: Did not see the definitions of “complete resolution” OR “PARTIAL RESOLUTION IN THE Methodology. Please address.
Discussion:
This section is too long, there is a significant amount of unnecessary material. Shorten it by at least 1/3. For example: delete paragraph on co-infections, etc…also section on diagnosis is too long and not relevant to this manuscript.
Line 286” what is meant by “funguses”. This is not a word. Maybe change to “cases”.
Line 411: What are the significant limitations of the study.
Comments on the Quality of English Language
as above, only minor corrections needed.
Author Response
Dear Editor,
We would like to thank you and the reviewers for evaluating our manuscript entitled: “Diagnosis, treatment and outcome of Coccidioidal osseous infections: A systematic review”. In the revised version of the manuscript, we tried to address adequately all points raised by the reviewers. Our responses to each point brought up are provided below. (Original comments in bold). Changes in the revised manuscript have been highlighted in red.
Reviewer 2
Materials and Methods:
Line 124-25: confusing, need to re-structure.
Authors’ response: This phrase has been re-structured so the meaning of this sentence is more clear now.
Line 128-130: Confusing, should be re-structured.
Authors’ response: This phrase has been re-structured so the meaning of this sentence is more clear now.
Results:
Line 176: change to “patients” not “patents”.
Authors’ response: This word has been corrected.
Line: authors state that the most common underlying condition was “ diabetes”. However, the most common condition per the Table 1 was steroids. This should be corrected. In addition to the “immunodeficiency state” which in text is 7.3%, however steroid use, which is an immunodeficiency per table is 22.6 %. Please address and correct.
Authors’ response: We agree with the reviewer that this may be confusing. By the phrase “underlying condition” we meant to refer to a specific disease such as diabetes or sarcoidosis. Corticosteroid intake cannot be considered a disease since the reason for corticosteroid intake and the duration of this intake was not mentioned in many studies. Therefore, some patients could be taking corticosteroids for an allergic reaction or for sciatica for a short period of time which could not be linked with an immunodeficient condition. Therefore, it was decided to not include corticosteroid intake as an immunodeficient condition. However, we agree with the reviewer that the manuscript need to be modified to make clear to the readers that the most common disease (and not condition which is a more vague term) was diabetes, to point out that corticosteroid intake was very common in these patients, and to state that the reason for corticosteroid intake and the duration of this intake was not mentioned in many studies
Line 193: what does a previous fungal infection” mean. Prior cocci vs others. If so what type of fungal infection?
Authors’ response: Previous fungal infections refer to an infection from another fungus other than Coccidioides spp., before the onset of the symptoms from the coccidioidal infection. Out of the 14 patients in which a previous fungal infection was reported, the causative pathogens were Candida spp. in 7 of them, and Aspergillus spp. in one of them. In the rest of these patients the causative pathogen was not mentioned. This is now mentioned in the revised manuscript.
Line 200-203: Not sure what the co-bacterial infections mean. Maybe just colonization from biopsy. Since it is very low rate, I would just delete.
Authors’ response: Based on the reviewer’s recommendation, this part of the manuscript regarding co-bacterial infections is now deleted.
Line 207-208: According to the table, the most common site of infection was the knee at 19.3%, not the spine. Please correct.
Authors’ response: We thank the reviewer for pointing out this mistake. Spine was indeed the most common location and by mistake the numbers of these two locations got mixed up. The correct frequency of spine was 19.3%, while the correct frequency of knee was 17.5%. This has been corrected in the revised manuscript.
Line 231-232: “followed” is not appropriate here. Consider change to “provided”.
Authors’ response: The word “followed” has been replaced by “provided” in the revised manuscript based on the reviewer’s recommendation.
Table 6 does not provide useful data. Consider deleting.
Authors’ response: Table 6 is now deleted based on the reviewer’s recommendation.
Line 246: Did not see the definitions of “complete resolution” OR “PARTIAL RESOLUTION IN THE Methodology. Please address.
Authors’ response: We agree with the reviewer that this is valuable information to the readers. Complete resolution was defined as complete clinical improvement without any clinical signs of infection, and partial resolution was defined as incomplete clinical improvement with partial resolution of the clinical signs of infection, with or without radiological findings of partial resolution of the infection. This is now mentioned in the Methodology section of the revised manuscript.
Discussion:
This section is too long, there is a significant amount of unnecessary material. Shorten it by at least 1/3. For example: delete paragraph on co-infections, etc…also section on diagnosis is too long and not relevant to this manuscript.
Authors’ response: The discussion section has been shortened by approximately 1/3 based on the reviewer’s recommendation.
Line 286” what is meant by “funguses”. This is not a word. Maybe change to “cases”.
Authors’ response: This part of the original manuscript has been deleted from the revised version.
Line 411: What are the significant limitations of the study.
Authors’ response: The current review possesses certain limitations. The most significant limitation is that most cases lack descriptions of antifungal agent dosages, drug serum levels, minimum inhibitory concentrations, and associated side effects. Another significant limitation is that a formal meta-analysis of the extracted data was not feasible, since there was high heterogeneity in the collected data; therefore, a limited analysis was performed mainly regarding the association between the final outcome and the type of treatment. The most significant limitations of our study are now clearly stated in the limitation paragraph of the revised manuscript.
On behalf of all the authors,
Yours sincerely,
Andreas G. Tsantes, MD
Laboratory of Haematology and Blood Bank Unit, "Attiko" Hospital, School of
Medicine, National and Kapodistrian University of Athens, Athens, Greece
Round 2
Reviewer 1 Report
Comments and Suggestions for Authors
The authors implemented all suggested adjustments and I recommend publishing the manuscript.
Reviewer 2 Report
Comments and Suggestions for Authors
No further recommendations. Agree to accept as is.